# In-the-Wild Affect Analysis of Children with ASD Using Heart Rate

**DOI:** 10.3390/s23146572

**Published:** 2023-07-21

**Authors:** Kamran Ali, Sachin Shah, Charles E. Hughes

**Affiliations:** 1Synthetic Reality Lab, Department of Computer Science, University of Central Florida, Orlando, FL 32816, USA; shah2022@umd.edu (S.S.); charles.hughes@ucf.edu (C.E.H.); 2Department of Computer Science, University of Maryland, College Park, MD 20742, USA

**Keywords:** autism spectrum disorder (ASD), emotion recognition, heart rate, smart bracelet, wearable

## Abstract

Recognizing the affective state of children with autism spectrum disorder (ASD) in real-world settings poses challenges due to the varying head poses, illumination levels, occlusion and a lack of datasets annotated with emotions in in-the-wild scenarios. Understanding the emotional state of children with ASD is crucial for providing personalized interventions and support. Existing methods often rely on controlled lab environments, limiting their applicability to real-world scenarios. Hence, a framework that enables the recognition of affective states in children with ASD in uncontrolled settings is needed. This paper presents a framework for recognizing the affective state of children with ASD in an in-the-wild setting using heart rate (HR) information. More specifically, an algorithm is developed that can classify a participant’s emotion as positive, negative, or neutral by analyzing the heart rate signal acquired from a smartwatch. The heart rate data are obtained in real time using a smartwatch application while the child learns to code a robot and interacts with an avatar. The avatar assists the child in developing communication skills and programming the robot. In this paper, we also present a semi-automated annotation technique based on facial expression recognition for the heart rate data. The HR signal is analyzed to extract features that capture the emotional state of the child. Additionally, in this paper, the performance of a raw HR-signal-based emotion classification algorithm is compared with a classification approach based on features extracted from HR signals using discrete wavelet transform (DWT). The experimental results demonstrate that the proposed method achieves comparable performance to state-of-the-art HR-based emotion recognition techniques, despite being conducted in an uncontrolled setting rather than a controlled lab environment. The framework presented in this paper contributes to the real-world affect analysis of children with ASD using HR information. By enabling emotion recognition in uncontrolled settings, this approach has the potential to improve the monitoring and understanding of the emotional well-being of children with ASD in their daily lives.

## 1. Introduction

Autism spectrum disorder (ASD) is a neurodevelopmental condition that limits social and emotional skills, and as a result, the ability of children suffering from ASD to interact and communicate is negatively influenced. The Centers for Disease Control and Prevention (CDC) reports that 1 in every 36 children in the US is diagnosed with ASD. It can be difficult to recognize the emotions of individuals with ASD, therefore making it hard to infer their affective state during an interaction. However, new technological advancements have proven to be effective in understanding the emotional state of children with ASD.

In recent years, wearable devices have been used to recognize emotions, detect stress levels, and prevent accidents using behavioral parameters or physiological signals [1]. The low cost and wide availability of wearable devices such as smartwatches have introduced tremendous possibilities for research in affect analysis using physiological signals. An advantage of a wearable device, such as a smartwatch, is its ease of use in real-time emotion recognition systems. There are many physiological signals that can be used for emotion recognition, but heart rate is relatively easy to collect using wearable devices such as a smartwatch, bracelet, chest belt, or headset. Nowadays, many manufacturers have marketed smartwatches that can monitor heart rate employing photoplethysmography (PPG) sensors or electrocardiograph (ECG) electrodes. Heart rate sensors in devices like the Samsung Galaxy Watch, Apple Watch, Polar, Fitbit, and Xiaomi provide a reliable instrument for heart-rate-based emotion recognition. Another significant aspect of using heart rate signals for affect recognition is its direct linkage with the human endocrine system and the autonomic nervous system. Thus, a more objective and accurate affective state of an individual can be acquired by using heart rate information. In this work, the Samsung Galaxy Watch 3 is employed to acquire the heart rate signal, as it is more comfortable to wear for participants with ASD compared to wearing a chest belt or a headset. This is especially important for those who are hypersensitive, a common experience of those with ASD.

Previous research shows that heart rate changes with emotions. In Ekman et al. [2] showed that heart rate had unique responses to different affective states. It was found that heart rate increased during the affective states of anger and fear, and decreased in a state of disgust. Britton et al. revealed that heart rate during a happy state is lower than heart rate during neutral emotion [3]. Similarly, Valderas et al. found unique heart rate responses when subjects were experiencing relaxed and fearful emotions [4]. Valdera’s experiments showed that the average heart rate is lower in a happy mood as compared to a sad mood.

Similarly, the field of robotics is opening many doors to innovate the treatment of individuals with ASD. Motivated by their deficiencies in social and emotional skills, some methods have employed social robots in interaction with children with ASD [5,6,7]. Promising results have been reported in the development of the social and emotional traits of children with ASD while supported by social robots [8]. Similarly, in Taylor et al. [9,10] taught children with intellectual disabilities coding skills using the Dash robot developed by Wonder Workshop. In this paper, an avatar is used in a virtual learning environment to assist children with autism (ASD) in improving communication skills while learning science, technology, engineering, and mathematics (STEM) skills. In particular, the child is given a challenge to program a robot, Dash™. Based on the progress and behavior of the child, the avatar provides varying levels of support so the student is successful in programming the robot.

Most emotion analysis studies of children with ASD use various stimuli to evoke emotions in a lab-controlled environment. The majority of these studies have employed pictures and videos to evoke emotions. However, in [11], Fadhil et al. report that pictures are not the proper stimuli for evoking emotions in children with ASD. Although most studies have used video stimuli, other research works have employed serious games [12] and computer-based intervention tools [13]. In this work, a human–avatar interaction is used in a natural environment where children with ASD learn to code a robot with the assistance of an avatar displayed on an iPad.

The compilation of ground truth labels from the captured data is challenging, laborious, and prone to human error. To tag the HR data according to emotions, many techniques have been used in the literature. For instance, in [14], the study participants use an Android application to record their emotions by self-reporting them in their free time. Similarly, in [15,16], the HR signals are labeled by synchronizing the HR data with the stimuli videos. Since the emotion label of the stimuli is known, the HR signals aligned with those stimuli are tagged accordingly. The problem with this tagging process is that it assumes the participants experience the emotion of the stimuli and that it is constant for all participants. However, in Lei et al. [17] reveal that individuals experience varied emotions to different stimuli.

The ground truth labeling process becomes even more challenging when the data are collected from participants with ASD [18]. For these participants, it is very difficult to accurately determine their internal affective state, and due to the deficits in communication skills in children with autism, the conventional methods for emotion labeling are difficult to apply [18,19]. In this paper, a semi-automatic emotion labeling technique is presented that leverages the full context of the environment in the form of videos captured during the interaction of the participant with the avatar. An off-the-shelf facial expression recognition (FER) algorithm, TER-GAN [20], is employed to produce an initial label recommendation by applying FER on the video frames. Based on the emotion prediction confidence of the FER algorithm, a human with knowledge of the full context of the situation decides the final ground truth label. The FER algorithm classifies a video frame into seven classes, i.e., the six basic expressions of fear, anger, sadness, disgust, surprise, and happiness, and a neutral state. Similar to [14], these emotions are clustered into three classes: neutral (neutral), negative (fear, anger, sadness, and disgust), and positive (happiness). After tagging the children–avatar interaction videos, the classical HR and video synchronization labeling technique is used to produce the ground truth emotion annotation of the HR signal.

After compiling the training and testing dataset, optimal features from the heart rate signal are then extracted and fed to the classifier for emotion recognition. A comparison between two different feature extraction techniques is also presented and experiments for intra-subject emotion categorization and inter-subject emotion recognition are performed. The main contributions of this paper are given below:To the best of our knowledge, this is the first paper that presents a wearable emotion recognition technique using heart rate information as the primary signal from a smart bracelet to classify the emotions of participants with ASD in real time.The dataset compiled for this study contains face videos and heart rate data collected in an in-the-wild set-up where the participants interact with an avatar to code a robot.A semi-automated heart rate data annotation technique based on facial expression recognition is presented.The performance of a raw HR-signal-based emotion classification algorithm is compared with a classification approach based on features extracted from HR signals using discrete wavelet transform.The experimental results demonstrate that the proposed method achieves a comparable performance to state-of-the-art HR-based emotion recognition techniques, despite being conducted in an uncontrolled setting rather than a controlled lab environment.

The following presents the overall structure of the paper. The Section 2 of this paper presents the related work in the domain of emotion recognition of kids with ASD. The Section 3 describes the methods and the experimental details such as the demography of participants, the in-the-wild real-time learning environment, the interaction of a child with an avatar, the semi-automatic emotion labeling process, the feature extraction techniques, and the emotion recognition step. The Section 4 presents the results of the experiments and discusses and compares the results with the state-of-the-art emotion recognition techniques using heart rate information. The Section 5 presents the conclusions, and the Section 6 is about the limitations and the future research work.

## 2. Related Work

Many techniques have been proposed in the last few decades to recognize human emotions employing different modalities such as facial expressions [21,22,23], speech signals [24,25], and physiological signals [26]. Here, the primary objective of the application of emotion recognition is to aid either an automated system or a human-in-the-loop with regard to a participant’s emotions during interactions within some scenario (typically a learning environment). Each control system (automated or human) uses these emotional data to direct the interactions of virtual characters or even to alter the environment during an experience.

Emotion recognition has also been applied by many researchers to improve the interaction of children with ASD and social robots [18,27,28,29]. Different modalities, such as facial expressions [29,30,31,32,33,34,35,36,37,38,39], body posture [30,40], gestures [41], skin conductance [11,19], respiration [42], and temperature [43], have been used to perform emotion analysis for children with ASD. Heart rate is also used to recognize the emotions of children with ASD, but these methods use HR as an auxiliary signal that is combined with other modalities such as skin conductance [19] or with body posture [30]. In this paper, an emotion classification technique is developed that uses HR as the primary signal.

Table 1 summarizes the studies on the emotion recognition of participants with ASD. Many of these techniques are complex and not suitable for applications in the real world. In this paper, an emotion classification technique is presented that uses HR as the primary signal leveraging a wearable smartwatch.

## 3. Methods

### 3.1. Subject Information

A total of nine children (6 male and 3 female), aged 8 to 11 years old, who met the criteria for ASD were recruited for this paper. Written parental consent and student assent were acquired from each child prior to participation. All of the procedures were approved, and both university and school district Institutional Review Board (IRB) approval for the study were obtained. All participants in our studies were clinically diagnosed with ASD in their school setting, and it was required that their primary language be English. Further documentation beyond that was not requested as they were receiving services under the Individualized Education Program with the label ASD.

### 3.2. Interaction of Children with Avatar

Each child is given a task to program a robot, and the avatar interacts with the child to assist in completing this task. During this process, the children will not only learn STEM skills but will also develop a relationship with the virtual avatar through communication. The avatar interacts with the child using an iPad. The time duration of these sessions varies depending on the speed of task completion by the children. Table 2 shows the time of interaction of each child with the avatar to complete the given task.

Due to the nature of the task and the interaction with the avatar, the children experience different emotions at various stages of the session. For instance, the child often becomes happy or surprised when the steps to program the robot are completed. Similarly, the child often feels sad or angry when the robot Dash fails to move based on the child’s intent. The videos of these sessions are recorded, where each frame contains the face of the participant, the window containing the avatar, the window showing the robot Dash, and the audio of the interaction between the child and the avatar, as shown in Figure 1. Each child wears a Samsung Galaxy 3 smartwatch that collects heart rate information. To transmit the heart rate data in raw form in real time, a smartwatch application was developed using the Tizen OS. Figure 2 shows the Samsung Galaxy watch with the Tizen-based real-time heart rate transmitting app, and the desktop app to receive the heart rate information.

### 3.3. Semi-Automated Emotion Annotation Process

The heart rate data are aligned with the videos of the children working with the avatar to complete the given task. To label the heart rate data, a semi-automated emotion classification algorithm based on facial expression recognition is developed. Figure 3 shows the flow diagram of the semi-automated emotion annotation process. During this annotation process, the off-the-shelf TER-GAN [20] FER model is leveraged, using added parameters to classify two more emotions, i.e., neutral and contempt, and fine-tuned on the in-the-wild AffectNet dataset [44]. Each video is divided into clips containing $n_{f}$ video frames, where $n_{f}$ is the multiplication of the frame rate of the video and the time segment corresponding to the window size. The frame rate of the videos in the dataset is 25 fps. For the window size of two seconds, the value of *n* is n=25×2=50. Then, the video clip is inputted to the FER model frame by frame to obtain a representative frame for the entire video clip, and the length of the video clip is synchronized with the transmission frequency of the heart rate sensor of the smartwatch. The representative frame is chosen based on two criteria: (1) the label of the representative frame should be the most frequent label, and (2) the prediction confidence of the frame should be the highest in the frequency list. After automatically obtaining the representative frame and its emotion label, the algorithm decides whether or not to employ a human annotator based on the confidence of the model predicting the emotion label. If the confidence value is lower than a threshold, then the human annotator steps in, and after analyzing the full context of the situation, the final label of the representative frame is assigned. Since the heart rate is aligned with the video data, the emotion label of the video is assigned to the corresponding heart rate data. In this paper, emotions are categorized into three classes: neutral, positive, and negative. Therefore, the heart rate data are clustered into these three groups.

### 3.4. Feature Extraction

The performance of the emotion classifier depends on the quality of the features extracted from the heart rate signal. As mentioned above, one of the main goals of this paper is to provide real-time support to a human puppeteer or automated system by classifying the emotion of a child based on heart rate information. Given this goal, the motivation is to avoid delays in the real-time processing of the heart rate signal. As such, experiments are conducted with three different time windows (five seconds, three seconds, and two seconds). Therefore, the heart rate data are obtained in the form of vectors:(1)$$V=(h_{\left(t-(n-1)\right)},h_{\left(t-(n-2)\right)},\ldots ,h_{t})$$

$h_{t}$ represents heart rate at time *t*.*n* corresponds to the length of the time window.

Features from the heart rate signal are then extracted for each time interval.

#### Discrete Wavelet Transform

Wavelet transform is widely used in signal processing applications to analyze signals in the time-frequency domain. This mathematical tool is also used in many research works to analyze heart rate data [8,45,46]. The discrete wavelet transform (DWT) is preferred over conventional signal analysis techniques in decomposing the waves into an optimal resolution, both in time and frequency. Thus, there is no requirement that the signal be stationary. Due to these desirable properties, DWT is frequently used in many research works to perform time-scale analysis, signal compression, and signal decomposition.

DWT filters decompose a signal into two bands at any particular level, i.e., approximations and detail bands of a signal. The approximations (*A*) correspond to the low-frequency components of the signal at a high resolution. The details (*D*) are high-frequency components of the signal at a lower resolution. During the sub-sampling process, the components of a signal are divided by 2 for multi-resolution analysis, as shown in Figure 4. The pre-processed heart rate data are inputted to the DWT decompositions. This multi-scale DWT decomposition is also called sub-band coding. The sub-sampling at every scale decomposes a signal into half the number of samples. Figure 5 shows the multiscale decomposition of a signal into sub-bands at various levels. In this paper, DWT features are extracted for emotion recognition by decomposing the heart rate signal using the Haar wavelets. The segments extracted from the signal using a window size of 2 s are fed to the DWT to extract the features. The Python PyWavelets [47] library is used to implement the DWT-based feature extraction algorithm.

### 3.5. Emotion Recognition

After extracting DWT features from the heart rate signal, three different classifiers are used to recognize emotions. SVM, KNN, and random forest (RF) [48] classifiers are used for intra-subject and inter-subject classification. Intra-subject emotion classification is performed when the heart rate data from the subject are acquired individually, and then the classifier is trained and tested on the same data, whereas in inter-subject emotion recognition, the heart rate data are collected from all participants rather than individually, and are used for the training and testing of the recognition modal. All of the experiments were performed using a ten-fold cross-validation method. For comparison purposes, similar to [16], experiments are also conducted using the heart rate data as the input feature to the classifiers. Figure 6 shows the heart rate signals of three different participants in the negative, positive, and neutral states.

The classification accuracy of the three classifiers is calculated using the following formula:(2)$$Accuracy=\frac{TP+TN}{TP+TN+FP+FN}$$
where TP represents true positive, TN denotes true negative, FP stands for false positive, and FN is the abbreviation of false negative [16].

## 4. Results and Discussion

Summary statistics of the heart rate data acquired during the interactions of children with the avatar are shown in Figure 7. Figure 7 shows the average heart rate, the minimum heart rate, and the maximum heart rate of all participants. As can be seen during the completion of tasks and the interaction with the avatar, the participants go through a range of heart rate activities. Figure 8 shows the maximum, minimum, and average beats per minute of all nine participants. The average heart rate of all nine participants is 96.8 BPM, while the maximum heart rate is 124 BPM (Participant 6) and the minimum heart rate is 62 BPM (Participant 1).

The emotion recognition results of both the intra-subject and the inter-subject data using DWT and heart rate features employing SVM, KNN, and RF classifiers are discussed in the following paragraphs. Experiments are performed using three different window sizes, and it is found that a window size of two seconds enhances the performance of the algorithm in terms of both accuracy and speed, which facilitates the real-time application of the emotion recognition technique.

The intra-subject classification accuracies of the three classifiers using DWT features are shown in Figure 9. In the case of SVM, the highest accuracy of 100% is obtained from Participant 6, and the lowest recognition accuracy of 40.1% is obtained from Participant 4. For KNN, Participant 6 obtains the highest accuracy of 100%, and Participant 3’s emotion recognition accuracy of 51.4% is the lowest. In the case of RF, the highest accuracy of 99.5% is obtained from Participant 6, and the lowest accuracy of 39.2% is obtained from Participant 9.

Similarly, the intra-subject classification accuracy of the three classifiers using HR data is shown in Figure 10. In the case of SVM, the highest accuracy of 100% is obtained from Participant 6, and the lowest recognition accuracy of 29.7% is obtained from Participant 1. In the case of KNN, Participant 5 obtains the highest accuracy of 100%, and Participant 2’s emotion recognition accuracy of 32.1% is the lowest. For RF, the highest accuracy of 99.2% is obtained from Participant 6, and the lowest accuracy of 35.6% is obtained from Participant 1.

The emotion recognition accuracy using the DWT features for inter-subject classification employing SVM, KNN, and RF is shown in Table 3. The highest emotion recognition accuracy among all three classifiers, with a window size of 2 s, is obtained with SVM, and the average accuracy of the ten-fold cross-validation is 39.8%. The recognition accuracy for the window size of three and five seconds is 39.8% and 39.9%, respectively, and since the window size of five has a less significant impact on the average accuracy with a lagging overhead, we set the window size to two seconds to facilitate the real-time application of the proposed method. The emotion classification accuracy produced by KNN and RF is 33.4% and 35.7%, respectively, using a window size of two seconds. Similarly, the highest classification accuracy of 38.1% is obtained using SVM with the heart rate signal as an input feature, while RF produces the lowest recognition accuracy of 31.9%, as shown in Table 4. Hence, this comparison indicates that a slightly better performance in emotion recognition can be achieved by using DWT-based features. Figure 11 shows the confusion matrix of experiments performed with DWT features and raw heart rate signal, while Table 4 shows the average precision, average recall, and F1 score of the experiments with the DWT features and the HR signal. Similarly, comparing the intra-subject and inter-subject recognition accuracy, it can be seen that the inter-subject emotion detection task is much more difficult than the intra-subject emotion classification due to the variation present in heart rate data for each individual.

### Comparison with Related Studies

The emotion recognition results are compared with the state-of-the-art HR-based emotion classification techniques in Table 5. As shown in the table, the highest recognition accuracy of 84% is obtained by [15], while the second highest accuracy of 79% is produced by the emotion recognition technique proposed in [15]. As reported in [16], the experimental protocol and details of the heart rate data used for the validation of these techniques are not explained in these papers. It is not known whether their emotion recognition algorithm is validated using the intra-subject HR data or the inter-subject HR data. Therefore, a better comparison of the classification accuracy of the technique presented in this paper can be performed by comparing the intra-subject recognition accuracy of 100% with the intra-subject classification accuracy in [16], which is also 100%. Similarly, the inter-subject recognition accuracy of the technique proposed in this paper is comparable to the inter-subject classification accuracy of the technique proposed in [16], despite the fact that the algorithm developed in this paper is trained and validated with the in-the-wild HR dataset obtained during the real-time interaction of the participant with an avatar without well-defined external stimuli and a constant lab environment. Another reason for the slightly lower recognition accuracy of the technique presented in this paper is that the number of participants in this study is nine, while the emotion recognition algorithm in [16] was trained and tested using a dataset of twenty participants. Note that it is common to have small samples when working with children diagnosed to be on the spectrum versus other populations.

## 5. Conclusions

The main objective of this paper is to develop a real-time heart-rate-based emotion classification technique to recognize the affective state of children with ASD while they interact with an avatar in an in-the-wild setting as opposed to a lab-controlled environment. A semi-automated emotion annotation technique based on facial expression recognition is presented for tagging the heart rate signal. The emotion labels obtained from the proposed tagging method are then grouped into three clusters: positive, negative, and neutral emotions. To classify the affective mood of a child with ASD into three emotional states, two sets of features are extracted from the HR signal using a window size of two seconds, and the effectiveness of these two sets of features is evaluated on three classifiers, namely, SVM, KNN, and RF. Two types of HR datasets are also compiled, i.e., an intra-subject dataset and an inter-subject dataset. The experimental results show that the classification accuracy obtained by extracting DWT and HR features from the intra-subject dataset is higher than the recognition accuracy of the inter-subject HR dataset. The experiments performed using the inter-subject HR dataset produce the highest emotion recognition accuracy by using the DWT features with SVM, which is comparable to the state-of-the-art inter-subject HR-based emotion classification technique. The variation in heart rate due to individual differences present in the inter-subject dataset contributes to lower recognition accuracy, as observed when using other HR-based emotion recognition techniques.

## 6. Limitations and Future Work

The semi-automated heart rate signal annotation technique involves human interference when the face video frames contain head poses that vary too far from the frontal face position. Similarly, the quality of the heart rate signal deteriorates with too frequent hand movements due to the addition of movement noise.

In our future work, the heart rate signal will be fused with the facial expression information using multi-modal deep-learning-based emotion representation techniques. The fusion of eye gaze and heart rate information for multi-modal emotion recognition will also be included in this extended work.

## Figures and Tables

**Figure 1 sensors-23-06572-f001:**
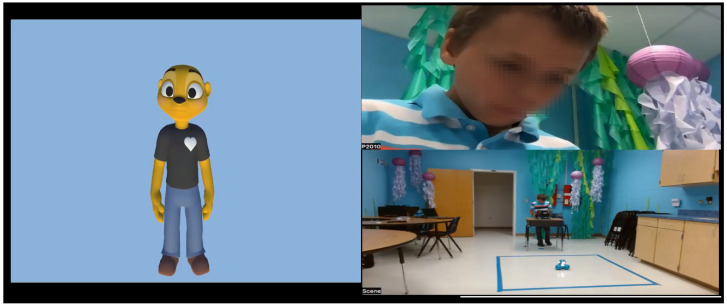
The experimental set-up for the child–avatar interaction.

**Figure 2 sensors-23-06572-f002:**
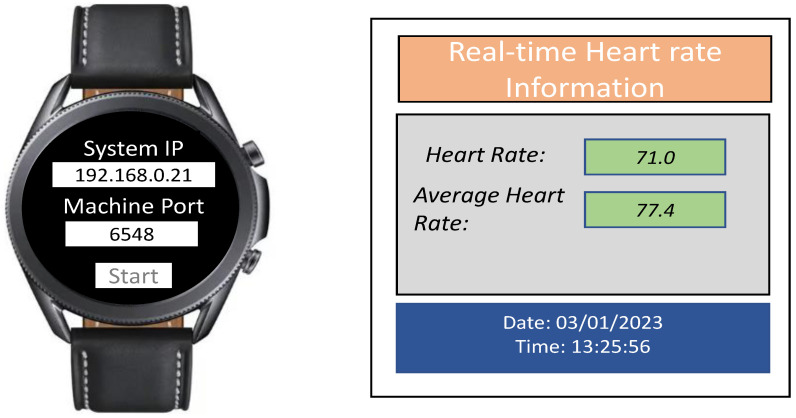
The Tizen-based real-time heart rate transmitting app and the desktop app to receive the heart rate information.

**Figure 3 sensors-23-06572-f003:**
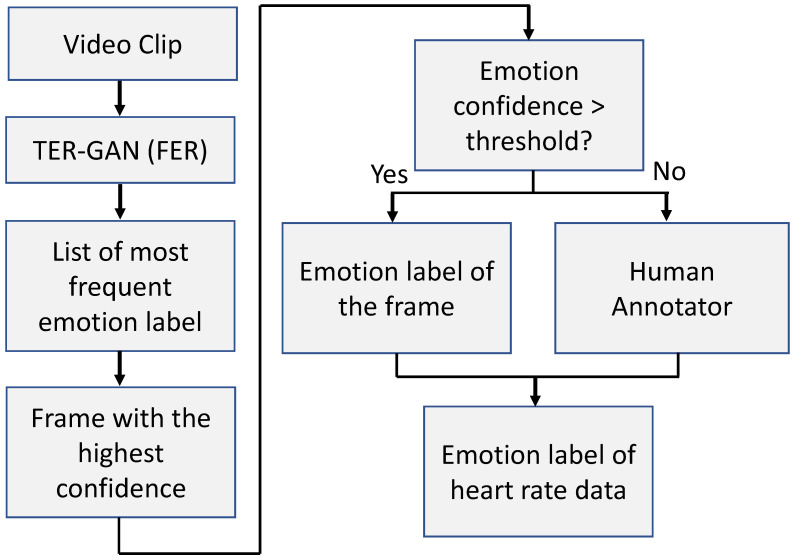
The flow diagram of the semi-automated emotion annotation process.

**Figure 4 sensors-23-06572-f004:**
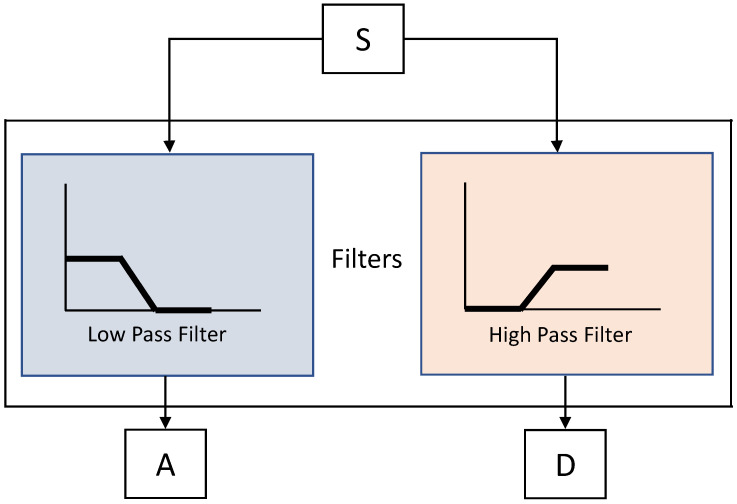
The low-pass and high-pass filtering of the DWT.

**Figure 5 sensors-23-06572-f005:**
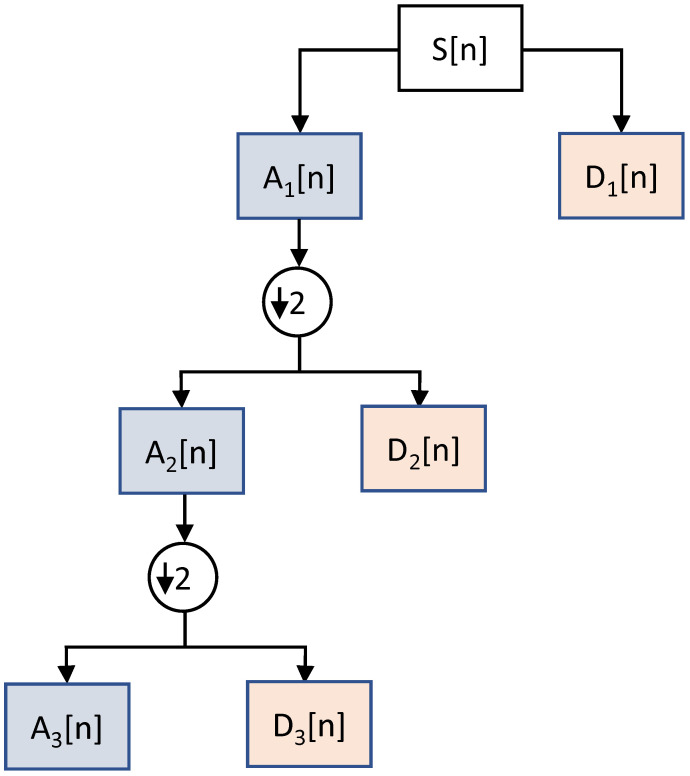
Discrete wavelet transform sub-band coding.

**Figure 6 sensors-23-06572-f006:**
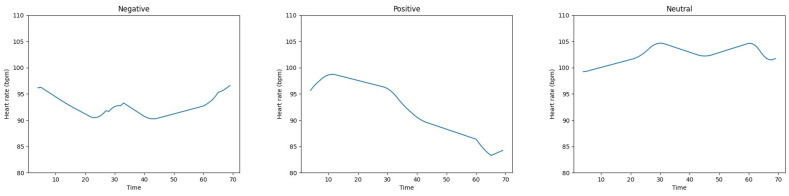
The heart rate signals of three different participants in the negative, positive, and neutral states.

**Figure 7 sensors-23-06572-f007:**
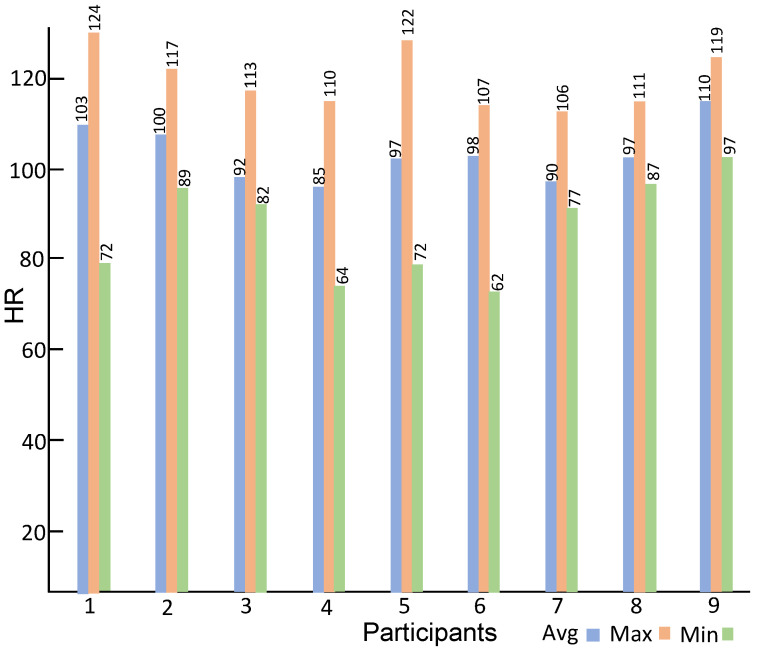
Average, minimum, and maximum heart rate of all participants.

**Figure 8 sensors-23-06572-f008:**
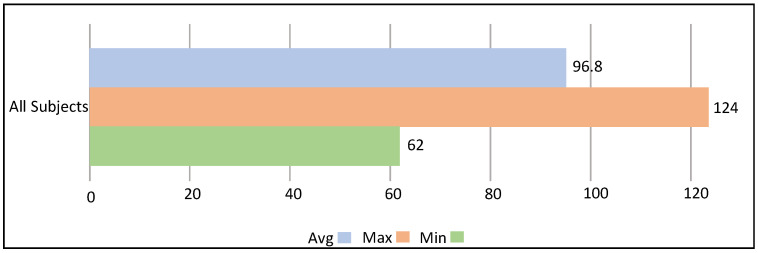
The average, minimum, and maximum heart rate of all participants collectively.

**Figure 9 sensors-23-06572-f009:**
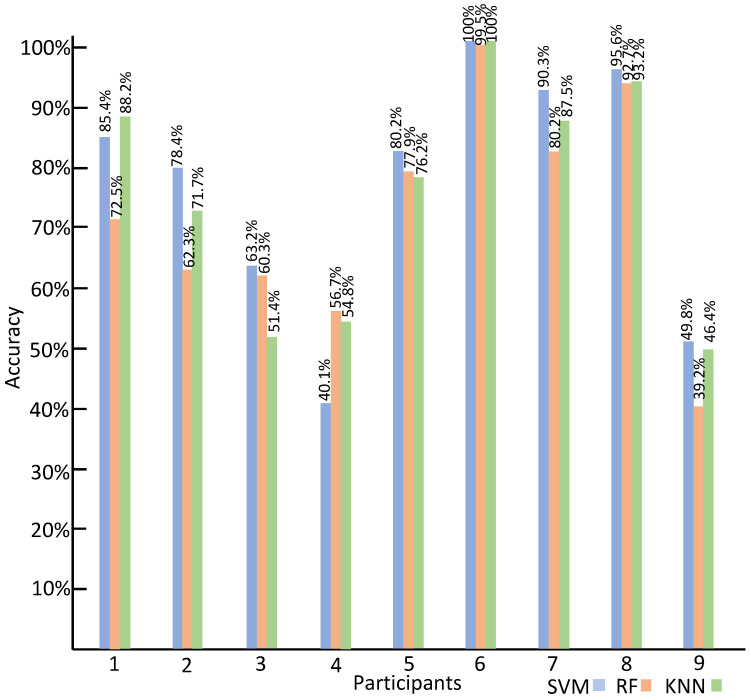
The intra-subject classification accuracy using DWT features.

**Figure 10 sensors-23-06572-f010:**
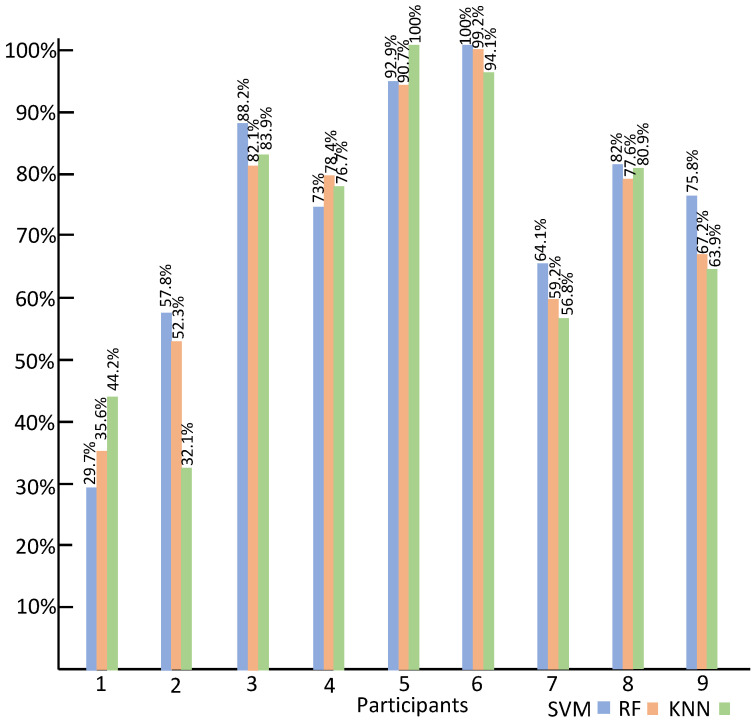
The intra-subject classification accuracy using HR.

**Figure 11 sensors-23-06572-f011:**
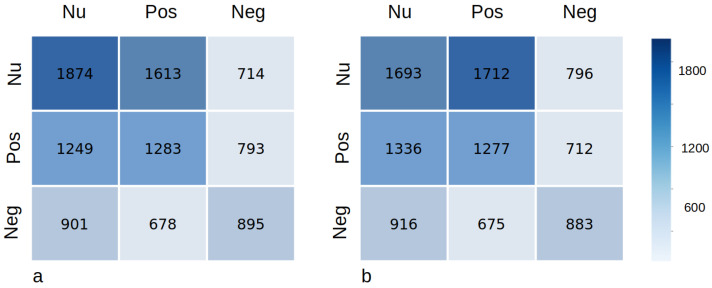
(**a**). The confusion matrix of the experiment performed with DWT features. (**b**). The confusion matrix of the experiment performed with the raw heart rate signal. ‘Neg’ stands for the negative class, ‘Pos’ represents the positive class, and ‘Nu’ stands for the neutral class.

**Table 1 sensors-23-06572-t001:** Summary of research on emotion recognition of ASD participants using various sensors.

Ref.	Related Work	Signal Type	Subject Number	Stimulation Materials	Performance
[39]	Grossard et al.	Video	36	Imitation of facial expressions of an avatar presented on the screen	Accuracy: 66.43% (neutral, happy, sad, angry)
[32]	Coco et al.	Video	5	Video	Entropy score: (happiness: 1776, fear: 1574, sadness: 1644)
[40]	Marinoiu et al.	Body posture videos	7	Robot-assisted therapy sessions	RMSE: (valence: 0.099, arousal: 0.107)
[41]	Kumar et al.	Gesture videos	10	Unknown	F-Measure: (angry: 95.1%, fear: 99.1%, happy: 95.1%, neutral: 99.5%, sad: 93.7%)
[28]	Liu et al.	Skin conductance	4	Computer tasks	Accuracy: 82%
[42]	Sarabadani et al.	Respiration	15	Images	Accuracy: (low/positive vs. low/negative: 84.5% and high/positive vs. high negative: 78.1%)
[43]	Rusli et al.	Temperature (thermal imaging)	23	Video	Accuracy: 88%

**Table 2 sensors-23-06572-t002:** Duration of the completion of the given task by the participants (seconds).

P1	P2	P3	P4	P5	P6	P7	P8	P9
828 s	846 s	786 s	540 s	660 s	480 s	583 s	611 s	779 s

**Table 3 sensors-23-06572-t003:** The inter-subject classification accuracy of all participants using SVM, RF and KNN.

Features	SVM	RF	KNN
DWT features	39.8%	35.7%	33.4%
HR signal	38.1%	31.9%	36.7%

**Table 4 sensors-23-06572-t004:** The average precision, average recall and F1 Score of the experiments with DWT features and HR signal.

Features	Average Precision	Average Recall	F1 Score
DWT features	0.399	0.397	0.398
HR signal	0.382	0.381	0.381

**Table 5 sensors-23-06572-t005:** Comparison with other HR-based emotion recognition techniques.

Author	Participants	Stimuli	Classifer	No. Classes	Accuracy
Shu et al. [15]	25	China Emotional Video Stimuli (CEVS)	Gradient boosting decision tree	3	84%
Bulagang et al. [16]	20	Virtual reality (VR) 360° videos	SVM, KNN, RF	4	100% for intra-subject and 46.7% for inter-subject
Nguyen et al. [14]	5	Android application	SVM	3	79%
Ours	9	Real-time interaction with avatar	SVM, KNN, RF	3	100% for intra-subject and 39.8% for inter-subject

## Data Availability

The videos/frames showing the faces of participants will not be made public as the IRB requires that access to each individual’s images has explicit parental permission.

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
