# Peer review of "In-the-Wild Affect Analysis of Children with ASD Using Heart Rate"

_sensors, 2023, doi:10.3390/s23146572_

Round 1
Reviewer 1 Report
The manuscript presents a framework to recognize the affective state of children with Autism Spectrum Disorder (ASD) using Heart Rate (HR) information.
Results demonstrate that the proposed method produces a comparable performance with the state-of-the-art HR-based emotion recognition techniques without the constraints of a controlled environment.
I find the topic interesting and being worth of investigation and the document is well structured.
Although I propose the following comments/suggestions:
- Abstract should be better organized: problem, motivation, aim, methodology, main results, further impact of those results.
- Keywords should be in alphabetical order
- I strongly suggest authors from refraining using personal pronouns such as "we" and "our" throughout the text and I encourage them to write it in an impersonal form of writing.
- Authors do not discuss the limitation of SVM being able of only recognize 2 classes
I strongly suggest authors from refraining using personal pronouns such as "we" and "our" throughout the text and I encourage them to write it in an impersonal form of writing.
Author Response
Thank you for the suggestions and feedback.
Re to "Abstract should be better organized ..."
We have organized the abstract of the paper as having a problem, motivation, aim, methodology, main results, and further impact of those results according to your suggestion.
Re to "Keywords should be in alphabetical order"
Done
Re to "I strongly suggest authors from refraining using personal ..."
Done
Re to “Authors do not discuss the limitation of SVM being able of only recognize 2 classes”
We have used a multiclass SVM to classify an emotion into three classes: positive, neutral, and negative.
Reviewer 2 Report
We appreciate the authors' contribution to advancing the recognition of affective states in children with autism spectrum disorders (ASD), while considering that this version of the paper has the following issues that require further explanation:
1. As the facial expressions of children with autism are usually not expressive, is it reasonable to use facial expressions as the basis for ground truth labelling. Furthermore, the sample size involved in the study was not large, so why a semi-automatic strategy was used and what advantages this strategy would have compared to purely manual annotation.
2. What is the difference in method design to [16] in Table 3, and would the performance be better or worse than your method if the method in [16] was transferred directly to the dataset you built.
3. Why the emotional stimulation manner in this paper can be called "in the wild" compared to methods such as [15,16,37] in Table 3.
Author Response
Thank you for the suggestions and feedback.
Re. 1. In the literature, there are various papers that use facial expression recognition to recognize the emotions of participants with ASD, such as the techniques proposed in [1, 2, 3, 4, 5, 6] in the references below.
The experiments are conducted on the video data of nine participants, and each participant has on average 15 videos, each video is of length 15 mins in our dataset. Our emotion recognition algorithm is part of a fully automated web-based WAVE platform [7] that is used by users in real-time. Therefore, the dataset is growing with usage, and the acquired dataset will be used to further improve the accuracy of the emotion recognition algorithm. Having a purely manual annotation scheme will hinder the continual development and improvement of the platform.
Re. 2. The main difference between our technique and the method proposed in [16] is that we extract HR features by employing Discrete Wavelet Transform (DWT), while the algorithm proposed in [16] uses raw HR signal as an input to the classifiers. To compare our DWT-based features with the raw HR signal of [16] on our dataset, we have conducted two sets of experiments, i.e., an inter-subject classification and an intra-subject classification experiment. As can be seen in the paper, our DWT-based heart rate features outperform the raw HR signal and boost the best classifier accuracy of SVM by 1.7%. While in the case of the intra-subject experiments, both of these features produced the best accuracy of 100% for Participant 6 using SVM as a classifier.
Re. 3. The videos of the participants in our dataset are not collected in a lab-controlled environment. The kids with ASD in our studies, interact with the avatar in a real-time setup with varying illumination, sometimes non-frontal head-poses, and with their hands and their hair occluding their faces. While the datasets collected in [15, 16, 37] are compiled in lab-controlled scenarios, as mentioned in their papers. Also, the emotional stimulation in our case is a response to the situation, and the avatar is not used to stimulate an emotion, but rather to react to the emotion of the participants.
References:
[1]. Gay, Valerie, Peter Leijdekkers, and Frederick Wong. "Using sensors and facial expression recognition to personalize emotion learning for autistic children." Stud. Health Technol. Inform 189 (2013): 71-76.
[2]. Leo, Marco, Marco Del Coco, Pierluigi Carcagni, Cosimo Distante, Massimo Bernava, Giovanni Pioggia, and Giuseppe Palestra. "Automatic emotion recognition in robot-children interaction for ASD treatment." In Proceedings of the IEEE International Conference on Computer Vision Workshops, pp. 145-153. 2015.
[3]. Silva, Vinícius, Filomena Soares, João Sena Esteves, Cristina P. Santos, and Ana Paula Pereira. "Fostering emotion recognition in children with autism spectrum disorder." Multimodal Technologies and Interaction 5, no. 10 (2021): 57.
[4]. Ghorbandaei Pour, Ali, Alireza Taheri, Minoo Alemi, and Ali Meghdari. "Human–robot facial expression reciprocal interaction platform: case studies on children with autism." International Journal of Social Robotics 10 (2018): 179-198.
[5]. Doyran, Metehan, Batıkan Türkmen, Eda Aydın Oktay, Sibel Halfon, and Albert Ali Salah. "Video and text-based affect analysis of children in play therapy." In 2019 International Conference on Multimodal Interaction, pp. 26-34. 2019.
[6]. Li, Jicheng, Anjana Bhat, and Roghayeh Barmaki. "A two-stage multi-modal affect analysis framework for children with autism spectrum disorder." arXiv preprint arXiv:2106.09199 (2021).
[7]. Shah, Sachin, Kamran Ali, Lisa Dieker, and Charles Hughes. "WAVE: a web-based platform for delivering knowledge-driven virtual experiences." IEEE Computer Graphics and Applications 43, no. 3 (2023): 54-60.
Reviewer 3 Report
The manuscript "In the wild Affect Analysis of Children with ASD using Heart rate" by Ali et al. is devoted to developing a method for mood recognition in children with ASD.
The work can be of interest to a broad scientific audience. However, in its current form, it is far from publication readiness. Moreover, assessing its scientific merits in its current form is impossible.
The basis for my assessment is the following:
1. Line 112: The authors stated, "The introduction section talks about the background, motivation, and main contributions of this paper." However, there is no discussion of the paper's contribution in the Introduction. For me, the contribution is unclear. The authors used X and Y to do Z, so what?
2. The whole article is structured inappropriately:
a. Introduction: It is more a free text narrative rather than an Introduction to the scientific paper. It comes in waves about each feature of the study. For example, the discussion of wearable devices for emotion recognition culminated with "In this work, we employ Samsung Galaxy Watch 3 to acquire the heart rate signal, as it is more comfortable to wear for participants having ASD compared to wearing a chest belt or a headset." After that, the authors similarly discussed stimuli and ground truth. The whole Introduction needs to be restructured. In particular, details of methodologies used by authors should be placed in the Methods.
b. There are no Methods. Instead, experimental methods are scattered through the Introduction and parts 2-5.
3. Implementation details are not sufficient. For example
a. From Eq.1, it is unclear what kind of input was used. What does HR data mean? How is this data spaced in time? For example, I understand the raw data from ECG or PPG sensor, which can be sampled a specific number of samples per second. From this information, the interbeat time period can be derived. This information is not equally spaced in time already.
b. Details of algorithm (e.g., DTW) implementations are missing. Did the authors use a standard software/package (e.g., Matlab) or code it themselves?
c. What exactly was fed into DTW? The whole interval, segments, windows?
d. What inputs were used in classifiers, particularly in the case of DTW
e. The window size is not analyzed properly. It is only stated that a window size of 2 is the best.
f. Line 152: "n video frames." What was the value of n? Is it the same n as in line 176, "n corresponds to the length of the time window"?
g. How data was split in training and test datasets?
4. Limitations and future work are missing
Minor shortcomings:
5. Line 59: The choice of words in "to help a human puppeteer" seems strange.
6. Table 1 should be referred closer to the paragraph about the technology used for emotion recognition.
7. Line 151: "in-the-wild AffectNet dataset" should be referenced properly
8. Eq. 2: The formula is incorrect. It should be TP+TN in the numerator.
9. Line 223: Abbreviations, like RF (Random Forest?), should be annotated on their first appearance.
10. In the case of the Random Forest classifier, it should be referenced
11. I don't see a value in Fig 8, 11, and 12. Each of them describes just 3 values
Author Response
Thank you for the suggestions and feedback.
Re. 1: We have summarized the contribution of our paper in bullet points now to make it more clear for the readers.
Re. 2: We have reorganized the structure of the paper to make it more clear, and have included a methods section as suggested by the reviewer.
Re. 3.a: As it is mentioned in the paper in line 174, the heart rate data is in the form of vectors, and this data is equally spaced in time. The Samsung Galaxy watch 3 has one channel PPG signal sampled at 50 Hz, and as you have mentioned from this information, the interbeat time period, in other words, the heart rate information can be derived.
Re. 3.b: We used the PyWavelets library to implement the DWT-based features extraction algorithm.
Re. 3.c: The segments extracted from the signal by using a window specified in the paper are fed to the DWT to extract the features.
Re. 3.d: DWT is only used to extract features, and as mentioned in the paper, three classifiers, namely, SVM, KNN, and RF are used for classification. The input to the classifiers is the DWT features and for the second experiment, the input is the raw HR signal.
Re. 3.e: The comparison of the performance of the three window sizes is given in the results and discussion section of the revised paper.
Re. 3.f: The frame rate of the videos in our dataset is 25 fps. For the window size of two seconds, the value of n is n = 25 * 2 = 50. While 'n' in line 176 corresponds to the length of the time window. We have updated the notations in the revised paper to avoid confusion.
Re. 3.g: As mentioned in the paper, line 204, we used a ten-fold cross-validation to perform all the experiments.
Re. 4: We have added the limitation and the future work sections in the revised paper.
Re. 5: The primary objective and application of the methods developed here are to aid either an automated system or a human-in-the-loop as regards a participant's emotions during interactions within some scenario (typically a learning environment). Each control system (automated or human) uses these emotional data to direct the interactions of virtual characters or even to alter the environment during an experience.
We have fixed the rest of the minor shortcomings pointed out by the reviewer, and we really appreciate the reviewer's suggestions and feedback.
Reviewer 4 Report
Dear Authors,
Please find the attached file for your reference. Please update the paper based on the comments and resubmit it.
Regards

Minor editing of English language required
Author Response
Thank you for the suggestions and feedback.
Re 6: In the paper, figure 9 shows the intra-subject classification accuracies using DWT features, figure 10 shows the intra-subject classification accuracies using HR, and Table 3 shows the inter-subject classification accuracies using DWT features and the inter-subject classification accuracies using HR.
We have addressed the rest of the suggestions provided by the reviewer in the revised paper.
Round 2
Reviewer 2 Report
The author answered my questions properly and I think the current version can be considered for publication.
The quality of the paper writing is qualified and you can consider enhancing the aesthetics of the figures in the paper.
Author Response
Thank you for the feedback. We have updated the figures in the revised version of the paper to enhance the aesthetics of the figures.
Reviewer 4 Report
Dear Authors,
Thank you for addressing all my comments and I don't have any further concerns about the manuscript.
Regards
Author Response
Thank you for the feedback and suggestions.